# LIMMT: Less is More for Motion Tracking

**Yu Guan** [* 1 2]   **Zekun Qi** [* 1 2]   **Chenghuai Lin** [2]   **Xuchuan Chen** [1 2]   **Wenyao Zhang** [3 2]
**Jilong Wang** [2]   **Xinqiang Yu** [2]   **He Wang** [4 2]   **Li Yi** [1 5]

## Abstract

We argue that high-quality motion data can steer tracking policies toward better optimization trajectories early in training. In this work, we introduce **LIMMT** (*Less Is More for Motion Tracking*). To our knowledge, this is the first *data-centric* study for physics-based humanoid motion tracking. We go beyond simply removing erroneous clips. We define motion data quality through three dimensions: **physics feasibility**, **diversity**, and **complexity**. We show that training with **under 3%** of AMASS yields better tracking performance than training with the full dataset. Extensive experiments and analyses validate the effectiveness of our framework.

## 1. Introduction

Motion tracking is one of the key aspects in Humanoid robot learning: it turns a library of reference motions into physically grounded behaviors that can be deployed as locomotion primitives, athletic skills, and compositional controllers. With the rapid scaling of motion capture (MoCap) corpora—from studio-quality datasets such as LaFAN1 (Harvey et al., 2020) and AMASS (Mahmood et al., 2019) to internet-scale collections reconstructed from videos via pose/mesh estimation (Lin et al., 2023; Zhang et al., 2025b; Fan et al., 2025)—it is tempting to believe that humanoid tracking will follow the "more data, better generalization" trajectory seen in vision and language.

However, physics-based imitation-RL has not consistently benefited from indiscriminate dataset scaling. In practice, state-of-the-art tracking systems (Peng et al., 2018; 2021; Chen et al., 2025; Ze et al., 2025b; Zhang et al., 2025d) remain using small and high-quality mocap data like

[1]Tsinghua University [2]Galbot [3]Shanghai Jiao Tong University [4]Peking University [5]Shanghai Qi Zhi Institute. Correspondence to: Li Yi <ericyi@mail.tsinghua.edu.cn>, He Wang <hewang@pku.edu.cn>.

*Proceedings of the 43$^{rd}$ International Conference on Machine Learning*, Seoul, South Korea. PMLR 306, 2026. Copyright 2026 by the author(s).

LaFAN1 (Harvey et al., 2020) and AMASS (Mahmood et al., 2019). Large in-the-wild corpora often introduce systematic artifacts—temporal jitter, foot sliding, ground penetration, implausible contacts, and other violations of rigid-body physics—that corrupt the imitation signal and can lead to brittle solutions or reward hacking (Shimada et al., 2020; Shin et al., 2024). At the same time, training over massive motion libraries is expensive: larger sets increase rollout diversity but also amplify the cost of reference sampling, curriculum design, and long-horizon optimization (Peng et al., 2018; 2021). As a result, "more motion data" can become *both noisier and harder to use*, and the learning system may converge early toward the wrong attractor from which later training struggles to recover.

This paper adopts a data-centric view and asks a more foundational question:

> *What makes motion data valuable for learning robust humanoid tracking policies?*

Our key insight for tracking is that data quality can dominate data quantity because it shapes the *initial optimization trajectory*. High-quality motion targets provide consistent, physically meaningful gradients that steer policy learning toward stable solutions early; low-quality or redundant motions can inject biased objectives and unstable gradients that waste computation and degrade final performance.

Crucially, "quality" is not merely the absence of broken clips. We argue that high-value MoCap for tracking should be characterized along **three complementary dimensions**: ❶ **physics feasibility**: whether a motion can be realized by a rigid-body humanoid without severe artifacts, ❷ **action diversity**: whether the library covers distinct behaviors rather than repeating frequent patterns, and ❸ **action complexity**: whether motions provide informative, dynamic supervision rather than near-stationary segments. Physics feasibility prevents training on impossible targets; diversity prevents diminishing returns from redundancy; complexity prioritizes motions that induce richer state-action visitation and stronger learning signals (e.g., higher velocity/acceleration energy) than static clips. Together, these dimensions explain why naive scaling often underdelivers: large corpora may contain many clips, but not many *useful* clips.

Based on this perspective, we present **LIMMT** (*Less Is More for Motion Tracking*), a principled curation framework that operationalizes feasibility, diversity, and complexity through **General Quality Selection (GQS)**. GQS is a **three-stage** pipeline designed to be simple, interpretable, and *plug-and-play* for existing trackers.

**Stage I: simulator-grounded physics filtering.** We replay each candidate motion in a rigid-body simulator and compute a composite feasibility score from physically interpretable indicators. Severe failure modes that are particularly destructive to imitation-RL (e.g., extended floating that indicates catastrophic reconstruction errors) are heavily penalized, followed by joint-velocity limit violations and ground penetration; rare or mild signals such as self-collision and jerk receive minimal weights. Motions below a threshold are discarded. This gating is essential: infeasible motions should not be allowed to influence downstream representation learning and sampling, because they can occupy and distort the embedding manifold.

**Stage II: semantic motion embedding for diversity.** On the remaining motions, we learn a continuous manifold that captures the *structural and rhythmic* similarity of behaviors, rather than superficial Euclidean pose differences. Concretely, we embed motions using **Harmonic Motion Embedding (HME)** implemented as a Periodic Autoencoder (Starke et al., 2022), producing phase-robust global descriptors. This representation provides a metric space where distances meaningfully reflect behavioral diversity (e.g., gait styles, athletic skills, transitions), enabling global subset selection to target coverage instead of redundancy.

**Stage III: diversity-aware, complexity-weighted subset selection.** Finally, we perform **Global Weighted FPS** over the learned embedding space to select a compact training library. Beyond geometric coverage, we compute a **complexity score** from physics-based motion intensity (e.g., kinetic energy and acceleration) and inject it into FPS as a small bias: the sampler primarily maximizes distance to the selected set (diversity), while preferring dynamically richer motions when candidates are comparable in distance. This yields a subset that is both broad in behavior and informative for learning.

The staged design of GQS encodes a second insight: **feasibility, diversity, and complexity must be addressed in the right order.** Filtering must come first; otherwise, physically broken motions can dominate the representation space and "win" diversity selection for the wrong reasons. Embedding learning must operate on feasible data to define a meaningful semantic manifold. Complexity weighting should come last; otherwise, high-energy artifacts may be over-selected. By respecting this hierarchy, LIMMT turns a large, noisy motion corpus into a compact, high-value training library that is cheaper to use and easier to learn from.

Empirically, LIMMT demonstrates a strong less-is-more effect across multiple tracking systems. On AMASS, GQS-curated subsets consistently improve tracking accuracy and success rates while using substantially less data. Remarkably, in some settings we find that training on **only 3% of AMASS** (requiring **under one hour** of data) can outperform training on the full dataset across all evaluated metrics, highlighting how removing harmful and redundant motions can be more effective than scaling. Moreover, these gains transfer in a *plug-and-play* manner across diverse trackers (e.g., Any2Track and TWIST-2), indicating that GQS improves the training signal rather than exploiting idiosyncrasies of a particular algorithm. Finally, we provide extensive ablations and analyses that isolate the contributions of each dimension and quantify how different physical failure modes influence policy learning. Our contributions include:

- **A data-centric perspective redefining "quality" over "quantity."** We demonstrate that physics feasibility, action diversity, and complexity—not dataset scale—are the decisive factors for robust tracking, challenging the blind scaling assumption in physics-based RL.

- **The General Quality Selection (GQS) framework.** We propose a hierarchical pipeline that first eliminates physically infeasible artifacts via simulator grounding, then maximizes behavioral coverage and dynamic richness using harmonic embeddings.

- **A "Less-Is-More" paradigm for efficient learning.** We show that training on just **3%** of curated data consistently outperforms using full corpora, providing a new, cost-effective blueprint for future motion data acquisition and humanoid policy training.

## 2. Related work

### 2.1. Learning Human Motion Tracking

The goal of human motion tracking is to deploy temporally consistent, robot-compatible whole-body trajectories from motion sequences to humanoid embodiments while respecting kinematic limits, contacts, and balance. Early steps toward physically grounded humanoid tracking (Luo et al., 2023a;b; Dallard et al., 2023; Zhang et al., 2024) primarily focus on imitating human motion. PHC (Luo et al., 2023a) learns a physics-based controller for virtual avatars by coupling motion imitation with stability under contact and joint constraints, thereby achieving long-horizon tracking in simulation. Building on this premise, recent systems progressively move toward real-world deployment. Early tracking works (Fu et al., 2024; He et al., 2024; Cheng et al., 2024; Ji et al., 2024; Yang et al., 2025; Xie et al., 2025; Li et al., 2025; Qin et al., 2023) develop full-stack pipelines that leverage large teleoperation or human-to-humanoid interfaces for whole-body control and autonomous skills (Zhang

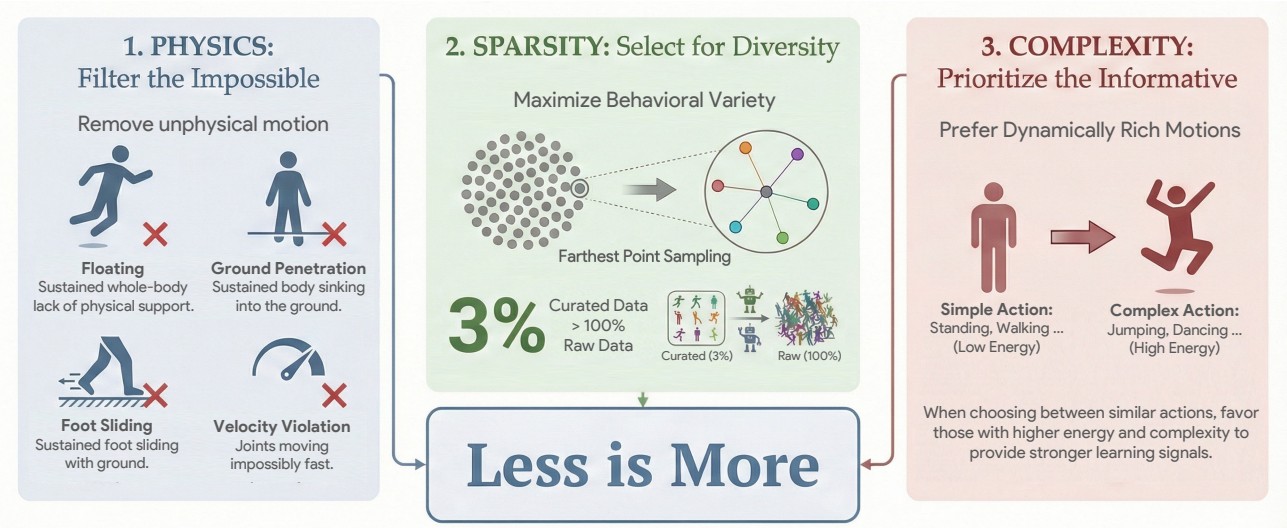

*Figure 1.* **The proposed GQS pipeline.** The framework operationalizes motion quality through three stages: filtering physically infeasible data, mapping motions to a semantic latent space, and selecting a subset via complexity-weighted sampling.

et al., 2026b) on specific humanoid platforms.

Recently, trackers have started to evolve toward stronger generalization capabilities. GMT (Chen et al., 2025) introduces a Mixture-of-Experts architecture with adaptive sampling to train a single, physically consistent policy capable of tracking more diverse motions. TWIST (Ze et al., 2025a) complements this effort from the teleoperation perspective(Zhang et al., 2025c), providing a real-time, human-in-the-loop whole-body imitation system for Unitree humanoids that emphasizes dynamic coordination and responsive control. UniTracker (Yin et al., 2025) advances this direction by learning a unified whole-body tracking policy through a CVAE-based teacher–student framework, enabling robust imitation of diverse human motions with improved generalization to unseen behaviors. SONIC (Luo et al., 2025) scales motion tracking into a universal whole-body control policy and further supports multi-modal and teleoperation-driven control interfaces. Humanoid-GPT (Qi et al., 2026) using a GPT-style Transformer trained on a billion-scale motion corpus for whole-body control, achieving zero-shot generalization to unseen motions.

### 2.2. Large-scale Motion Data

Large-scale motion datasets have become essential for learning generalizable human motion tracking. Early datasets (Harvey et al., 2020; Mahmood et al., 2019; Li et al., 2023) offered high-quality but studio-constrained motions, limiting diversity. With video-based reconstruction and large-scale synthetic generation, recent datasets greatly expand motion coverage, incorporating diverse activities, styles, and subjects with multimodal supervision (Lin et al., 2023; Zhang et al., 2025b; Fan et al., 2025; Mao et al., 2024).

More recently, (Lee et al., 2025) provides physically consistent motions with contact modeling, joint constraints, and reduced foot-sliding, offering stability benefits over purely kinematic sources. Together, these increasingly diverse and physically grounded datasets supply richer motion variety and stronger physical priors, forming key foundations for unified and robust human motion tracking systems (Kim et al., 2025; Qi et al., 2026; Xue et al., 2026).

## 3. Methodology

We propose **GQS** (General Quality Selection), a three-stage pipeline that transforms a large, noisy motion corpus into a compact, high-value training subset. As illustrated in Figure 1, GQS first filters physically infeasible motions, then learns a semantic embedding space for diversity measurement, and finally performs complexity-weighted sampling to select the training set.

### 3.1. Physics-based Feasibility Filtering

Raw motion capture datasets such as AMASS and MotionX++ are inherently kinematic and often violate the physical constraints of humanoid robots. Artifacts like prolonged aerial suspension, ground penetration, and joint velocity violations can lead to policy failure during sim-to-real transfer. To address this, we design a two-stage evaluation mechanism tailored for the target robot.

**Hard Constraints.** We first apply a binary validity check. A trajectory is discarded if its duration is less than 0.5 seconds, providing insufficient context for tracking, or if the joint velocity violation exceeds a safety margin of 0.05 rad/s, indicating mechanically impossible motions.

**Soft Scoring Function.** For valid trajectories, we compute a quality score $S_{phy}$ based on weighted penalties:

$$S_{phy}(\mathcal{T}) = 100 - \sum_i w_i \mathcal{L}_i \qquad (1)$$

The penalty terms $\mathcal{L}_i$ capture six distinct physical violation modes. **Floating** $\mathcal{L}_{float}$ detects non-physical suspension by applying a temporal convolution window over the foot-ground distance, measuring the ratio of frames where the robot is continuously airborne. **Ground Penetration** $\mathcal{L}_{pen}$ measures the average depth of mesh penetration into the floor. **Velocity Violation** $\mathcal{L}_{vel}$ captures the mean magnitude by which joint velocities exceed hardware limits. **Foot Sliding** $\mathcal{L}_{slide}$ penalizes horizontal foot velocity when the foot height is below 5cm. **Self-Collision** $\mathcal{L}_{col}$ penalizes collisions between robot links. **Jerk** $\mathcal{L}_{jerk}$ measures the rate of change of joint acceleration.

The weights $w_i$ are calibrated through a data-driven sensitivity analysis detailed in Sec. 4.6, where we measure each metric's impact on downstream policy performance. We retain trajectories with $S_{phy} \geq 90$, ensuring the dataset consists of physically executable motions.

### 3.2. Semantic Motion Embedding

To measure motion similarity in a semantic space rather than Euclidean joint space, we learn a continuous motion manifold using a **Periodic Autoencoder (PAE)**. Standard autoencoders often struggle to distinguish dynamically similar but temporally shifted motions. Our PAE addresses this by explicitly decomposing motion into periodic parameters.

The network processes a temporal window $X \in \mathbb{R}^{T \times D}$ with $T = 4.0s$, consisting of joint positions, velocities, and root velocities. Instead of encoding $X$ into a static latent vector, the encoder $E_\phi$ maps the input to dynamic parameters in a frequency domain: Amplitude $A$, Frequency $F$, Phase Shift $\phi$, and Offset $b$, each as a vector in $\mathbb{R}^k$ with $k = 8$. The latent trajectory is analytically reconstructed using a sinusoidal prior:

$$z_i(t) = A_i \sin(2\pi(F_i \cdot t + \phi_i)) + b_i, \quad i \in \{1, \ldots, k\} \quad (2)$$

The decoder $D_\psi$ then reconstructs the motion trajectory $\hat{X}$ from this explicit latent dynamics.

Unlike Variational Autoencoders that impose a probabilistic prior on the latent space, our PAE employs a purely deterministic mapping optimized solely via reconstruction loss. This ensures that learned embeddings faithfully preserve the physical scale and temporal frequency of motions without being distorted by regularization constraints. Consequently, the Euclidean distance in this manifold directly reflects physical discrepancies in motion intensity and pace.

**Global Embedding Extraction.** To perform global sampling, we require a time-invariant representation for each

---

**Algorithm 1** Global Weighted FPS

**Input:** Global Embeddings $\mathcal{Z}$, Complexity Scores $\mathcal{C}$, Target Count $K$, Weight $\alpha$
**Output:** Selected Indices $S$
Initialize $S \leftarrow \emptyset$, Candidate set $U \leftarrow \{1, \ldots, N\}$
*// Anchor Selection: start with the hardest motion*
$idx_{best} \leftarrow \arg\max_{i \in U} \mathcal{C}_i$
$S \leftarrow S \cup \{idx_{best}\}, U \leftarrow U \setminus \{idx_{best}\}$
Initialize distances $D[i] \leftarrow \|\mathbf{z}_i - \mathbf{z}_{best}\|, \forall i \in U$
*// Iterative Weighted Selection*
**while** $|S| < K$ **do**
   Normalize distances: $\hat{D} \leftarrow D / \max(D)$
   **for** each candidate $u \in U$ **do**
      $Score_u \leftarrow \alpha \cdot \hat{D}[u] + (1 - \alpha) \cdot \hat{\mathcal{C}}[u]$
   **end for**
   $next \leftarrow \arg\max_{u \in U} Score_u$
   $S \leftarrow S \cup \{next\}, U \leftarrow U \setminus \{next\}$
   Update distances: $D[i] \leftarrow \min(D[i], \|\mathbf{z}_i - \mathbf{z}_{next}\|)$
**end while**

---

variable-length motion sequence. We observe that the dynamic signature of a motion is primarily determined by its intensity $A$ and pace $F$, while phase $\phi$ and offset $b$ merely represent temporal alignment and pose bias. Therefore, for each window $w$ in a sequence, we construct a local descriptor $h_w = [A_w, F_w] \in \mathbb{R}^{2k}$. The final global embedding for a full trajectory is obtained by temporal averaging:

$$\mathbf{z}_{global} = \frac{1}{N} \sum_{w=1}^N [A_w, F_w] \qquad (3)$$

This yields a compact, phase-invariant embedding that provides a continuous metric space for global sampling.

### 3.3. Global Weighted FPS

While the PAE embeddings provide a continuous semantic manifold, the selection of training samples requires a strategy that goes beyond geometric coverage. We propose **Global Weighted FPS** based on a fundamental hypothesis: motions with higher dynamic complexity are inherently more challenging to track, thereby providing richer supervision signals and driving superior policy performance. Under this assumption, a dataset biased towards physically demanding skills should yield a more robust tracker than one dominated by static or simple motions.

**Motion Complexity Metric.** We quantify the training value of each motion via its physical attributes. The complexity $C(x)$ of a trajectory is defined as a weighted combination of kinetic energy and acceleration magnitude, representing the intensity and abruptness of movement:

$$C(x) = \frac{1}{T} \sum_{t=1}^T \left( \|\dot{q}_t\|_2^2 + \lambda \|\ddot{q}_t\|_2^2 \right) \qquad (4)$$

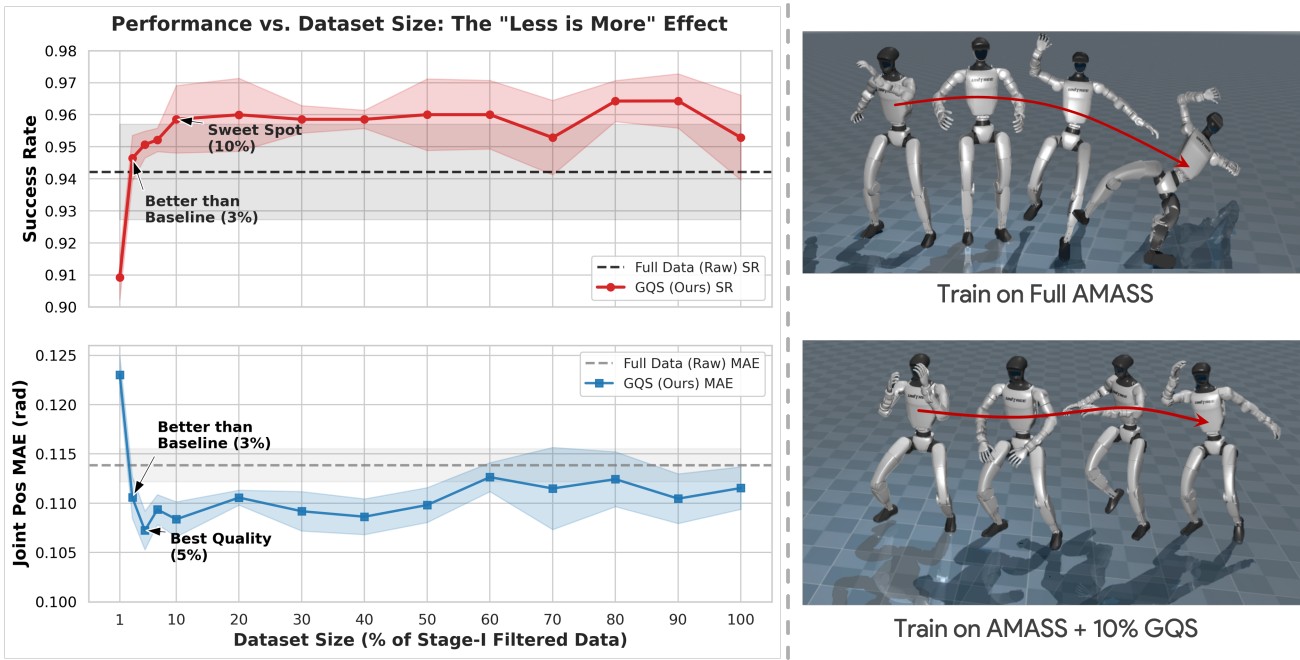

*Figure 2.* **Performance vs. Data Ratio.** The red line shows GQS Success Rate crossing the baseline at just 3% data. The blue line shows that tracking error peaks around 5%-10%. GQS significantly outperforms the Full Data baseline in both efficiency and quality.

where $\dot{q}_t$ and $\ddot{q}_t$ denote joint velocities and accelerations. We apply rank-based normalization to map $C(x)$ to the range $[0, 1]$, denoted as $\hat{C}(x)$.

**Sampling Strategy.** We initialize the selected set $S$ with the highest-complexity anchor to ensure the dataset is grounded in the most challenging demonstration. In each iteration, we select the candidate $u$ that maximizes a hybrid score:

$$\text{Score}(u) = \alpha \cdot \hat{D}(u, S) + (1 - \alpha) \cdot \hat{C}(u) \qquad (5)$$

where $\hat{D}(u, S)$ is the normalized distance to the nearest neighbor in the latent space. The hyperparameter $\alpha$ governs the interplay between spatial exploration and complexity maximization, which maintains the strong global exploration capability of standard FPS while introducing a physics-aware bias that favors dynamically richer motions when candidates are geometrically equidistant. This subtle integration of physical difficulty yields a better performance-to-data-ratio than purely geometric sampling.

## 4. Experiments

### 4.1. Experimental Setup

**Datasets.** We conduct our primary evaluations on the **AMASS** dataset (Mahmood et al., 2019), a large-scale motion capture corpus unifying 15 optical marker datasets. Following standard protocols, we use the official train/test split with approximately 14K training clips and 140 test trajectories, including mirrored versions. To verify generalizability, we also perform experiments on the **PHUMA**

dataset (Lee et al., 2025), a recently large-scale physically-grounded motion dataset with its official train/test split.

**Environment and Task.** We conduct experiments on two physics simulation platforms to validate generalizability: MJX for Any2Track and Isaac Lab for TWIST2, both enabling massive parallel training on GPUs. The robot platform is the Unitree G1. The downstream task is Universal Motion Tracking: given an unseen reference trajectory from the test set, the policy must control the humanoid to reproduce the motion as closely as possible without falling.

**Baselines and Metrics.** We establish the Full Data setting, Random selection and PHC (Luo et al., 2023a) as our primary baseline, training on the complete unfiltered AMASS dataset. We report the success rate to measure the percentage of trajectories tracked to completion, and MPJPE, the Mean Per Joint Position Error in radians.

**Implementation Details.** We use two State-of-the-art tracking policy Any2Track (Zhang et al., 2025d) and TWIST2 (Ze et al., 2025b) as our baseline. We train all policies using PPO for $2 \times 10^9$ environment steps on $8\times$ NVIDIA RTX 4090 GPUs. We train 10 seeds for every setting and report the mean and standard deviation.

### 4.2. Main Results

We evaluate GQS across data ratios from 1% to 100% and compare against multiple baselines. Figure 2 visualizes the performance trends, and Table 1 details key results.

*Table 1.* **Main Results on AMASS.** Comparison of Success Rate, MPJPE, and MPKPE across different methods and data ratios. Physics Filter indicates Stage-I filtering, and FPS Ratio denotes the Stage-II subset ratio.

| Method | Physics Filter | FPS Ratio | Success Rate ↑ | MPJPE (rad) ↓ | MPKPE (mm) ↓ |
|---|---|---|---|---|---|
| Any2Track (Zhang et al., 2025d) | × | - | $0.942 \pm 0.011$ | $0.114 \pm 0.003$ | $39.24 \pm 1.12$ |
| Any2Track+Random | ✓ | Random **3%** | $0.838 \pm 0.018$ | $0.159 \pm 0.005$ | $158.76 \pm 14.34$ |
| Any2Track+PHC (Luo et al., 2023a) | ✓ | - | $0.948 \pm 0.009$ | $0.111 \pm 0.002$ | $36.18 \pm 0.98$ |
| Any2Track+**GQS** | ✓ | 100% | $0.954 \pm 0.011$ | $0.112 \pm 0.003$ | $34.12 \pm 0.95$ |
| Any2Track+**GQS** | ✓ | 10% | $\mathbf{0.959} \pm 0.010$ | $\mathbf{0.107} \pm 0.002$ | $30.15 \pm 0.81$ |
| Any2Track+**GQS** | ✓ | **3%** | $0.956 \pm 0.012$ | $0.108 \pm 0.002$ | $\mathbf{29.87} \pm 0.76$ |
| TWIST2 (Ze et al., 2025b) | × | - | $0.825 \pm 0.014$ | $0.099 \pm 0.003$ | $35.80 \pm 1.08$ |
| TWIST2+Random | ✓ | Random **3%** | $0.649 \pm 0.021$ | $0.177 \pm 0.006$ | $263.19 \pm 27.87$ |
| TWIST2+PHC (Luo et al., 2023a) | ✓ | - | $0.845 \pm 0.012$ | $0.096 \pm 0.002$ | $33.54 \pm 0.94$ |
| TWIST2+**GQS** | ✓ | 100% | $0.843 \pm 0.012$ | $0.094 \pm 0.003$ | $31.25 \pm 0.89$ |
| TWIST2+**GQS** | ✓ | 10% | $\mathbf{0.868} \pm 0.011$ | $\mathbf{0.084} \pm 0.002$ | $27.21 \pm 0.72$ |
| TWIST2+**GQS** | ✓ | **3%** | $0.861 \pm 0.013$ | $0.092 \pm 0.002$ | $\mathbf{27.09} \pm 0.68$ |

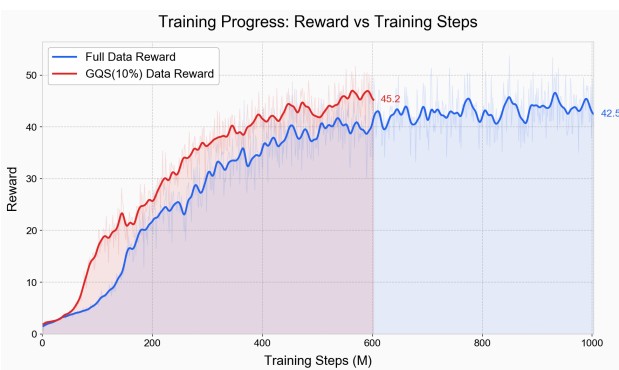

*Figure 3.* **Training Dynamics.** Learning curves comparing GQS 10% against Full Data. We achieve higher reward throughout training, not just at convergence, confirming that data curation improves the optimization trajectory from **early stages**.

**Random Subsampling Fails.** To isolate the effect of intelligent curation from mere data reduction, we include a Random 3% baseline that uniformly samples from the raw dataset. The results are striking: Random 3% causes catastrophic performance collapse, with Any2Track dropping to 83.8% SR and TWIST2 to just 64.9% SR. In contrast, GQS 3% *surpasses* the full-data baseline, achieving 95.6% and 86.1% SR respectively. This demonstrates that the "less is more" effect is not about using less data per se, but about using the *right* data.

**The Less-is-More Effect.** At just 3% data, GQS outperforms the Raw Full Data baseline across all metrics. Physics filtering alone at 100% scale already improves SR from 94.2% to 95.4%, confirming that removing infeasible motions helps. Further applying diversity-based selection yields even greater gains: GQS 10% reaches 95.9% SR, the optimal trade-off between efficiency and performance. The ceiling at 90% data offers only marginal improvement over 10% despite massive additional cost.

*Table 2.* **Component Ablation at 3% Data Ratio.** The full GQS framework achieves the best performance, demonstrating the synergistic benefit of all three components.

| Components | | | Metrics | |
|---|---|---|---|---|
| Physics | Sparsity | Complexity | Success Rate | MPJPE (rad) |
| × | ✓ | ✓ | $0.911 \pm 0.014$ | $0.1213 \pm 0.001$ |
| ✓ | × | ✓ | $0.934 \pm 0.009$ | $0.1197 \pm 0.003$ |
| ✓ | ✓ | × | $0.946 \pm 0.008$ | $0.1079 \pm 0.002$ |
| ✓ | ✓ | ✓ | $\mathbf{0.956} \pm 0.012$ | $\mathbf{0.1079} \pm 0.002$ |

**MPJPE as the Primary Metric.** As Success Rate approaches the 95% ceiling, MPJPE becomes more informative. GQS delivers **5%–15%** relative MPJPE reduction in a fully plug-and-play manner. For TWIST2, MPJPE improves from 0.099 to 0.084 rad, a 15% gain that directly translates to better motion fidelity in downstream applications.

### 4.3. Component Analysis

To evaluate the contribution of each component in GQS, we conduct ablation studies in the extreme low-data regime at 3% data ratio. This setting serves as a stress test where the quality of every selected motion is critical. Table 2 summarizes the results comparing the full GQS method against variants removing each component.

> *Physics Filtering is Critical.*

Removing the simulator-grounded physics filter leads to significant performance degradation, with Success Rate dropping from 95% to 91.1% and MPJPE worsening to 0.121 rad. This sharp decline confirms that embedding-based sampling inherently favors outliers, which without prior filtering are often physically infeasible artifacts. In the low-data regime, these toxic samples occupy valuable slots in the core set, severely hindering policy learning.

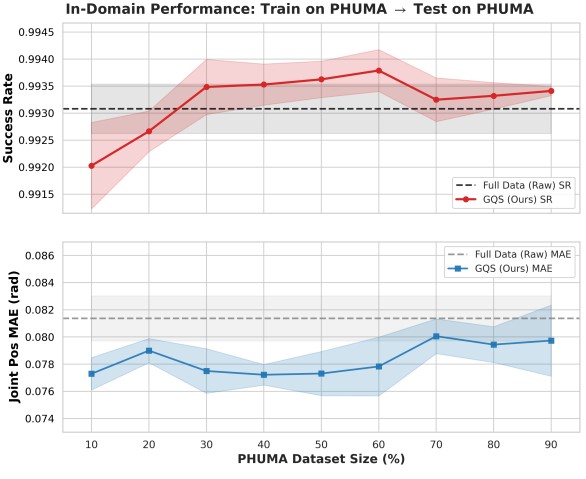

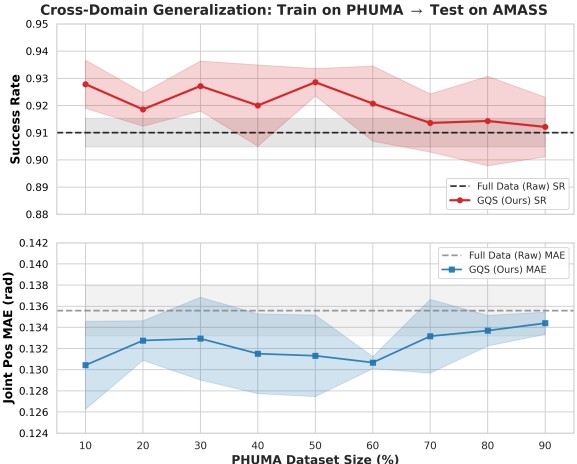

*(a)* **In-Domain Efficiency on PHUMA.**          *(b)* **Cross-Domain Transfer to AMASS.**

*Figure 4.* **Generalization Analysis on PHUMA.** In-domain evaluation shows the 10% subset achieves lower MPJPE than the full dataset. Cross-domain evaluation demonstrates the 10% subset significantly outperforms the full dataset when transferred to AMASS.

---

> *Diversity and Complexity Work Synergistically.*

Selecting motions purely based on complexity leads to suboptimal performance at 93.4% SR, indicating that covering the semantic manifold is the primary prerequisite. A curriculum focused solely on hard samples fails if it leaves large gaps in the behavior space. While vanilla FPS achieves competitive results at 94.6% SR, the full framework with complexity weighting achieves the best performance at 95.6%. This demonstrates that the three factors work synergistically: diversity ensures broad coverage, while complexity biases further refine selection to prioritize informative motions.

The value of GQS lies in its flexibility as an adaptive framework. It provides a unified approach where $\alpha$ acts as a domain-adaptive knob, favoring pure diversity for broad noisy datasets and shifting towards complexity mining for curated data or cross-domain generalization.

> *Does GQS Improve Early Optimization Trajectory?*

To verify whether data curation improves the optimization trajectory rather than just final performance, we track training dynamics throughout the $2 \times 10^9$ environment steps. Figure 3 compares the learning curves of GQS-curated data at 10% against the raw full dataset.

The results reveal that GQS-curated data achieves higher reward and lower tracking error from the early stages of training before 0.5B steps, confirming that curated data provides cleaner gradients that steer the policy toward better solutions early. The performance gap is maintained throughout training, indicating that the advantage reflects a fundamentally better optimization trajectory rather than merely faster convergence.

## 4.4. Cross-Dataset Generalization

To further validate the GQS system, we extend our evaluation to the PHUMA dataset (Lee et al., 2025). PHUMA is distinct from AMASS, with more curated and physically grounded filtering. This allows us to evaluate the value of stage II&III directly and prove that **physical is just part of the motion data quality, not all**. Besides, it provides a different data regime through both in-domain efficiency and cross-domain robustness experiments.

**In-Domain Precision.** On the held-out PHUMA test set, the Full Dataset baseline achieves a near-saturated Success Rate of 99.31%. Despite this challenging ceiling, GQS achieves consistently lower MPJPE across all evaluated data ratios from 10% to 90%, proving that our curation strategy universally improves tracking precision. Remarkably, GQS surpasses the performance ceiling using only 30% of the data, validating that GQS identifies a core subset that is both more precise and functionally superior to the full corpus.

**Cross-Domain Robustness.** When evaluated zero-shot on the unseen AMASS dataset, the 10% subset outperforms the Full Dataset in Success Rate with 92.8% vs 91.0%. This confirms that removing easy or redundant data prevents overfitting to source domain artifacts, thereby improving zero-shot ability. The complexity-biased selection acts as hard-negative mining to learn transferable dynamic features.

These results present a compelling case: GQS does not simply survive with less data but thrives. By curating a compact, complexity-rich dataset at 10%, we improve precision in-domain and robustness cross-domain, while reducing training costs by an order of magnitude.

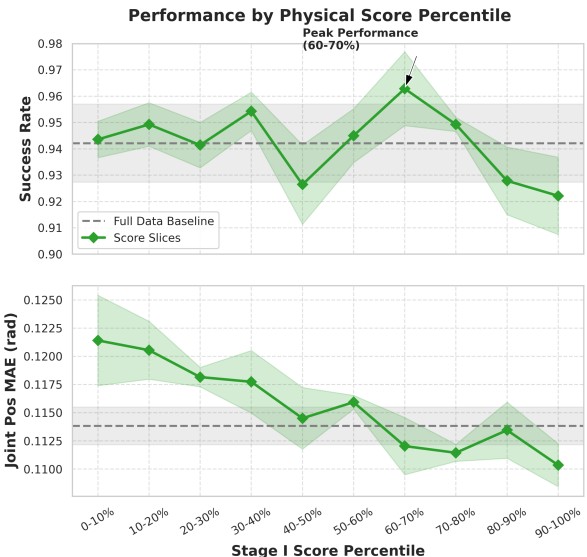

*Figure 5.* **Performance vs. Physical Score Percentile.** Performance drops at the tail but peaks at 60-70% rather than 0-10%, showing that physical filter alone doesn't determine training value.

### 4.5. Physical Score Analysis

The key difference from PHUMA is that we go beyond physical plausibility: we explicitly account for the effects of motion diversity and motion complexity on tracking performance. To verify that physics-only filtering is insufficient, we sorted all motions by physical scores in descending order, performing sorting within each cluster to avoid distribution shift. The data was divided into 10 percentile bins, with the 0-10% slice containing highest-scoring motions and 90-100% slice containing lowest-scoring ones. We trained separate policies on each slice controlling for data size.

Figure 5 reveals a critical non-monotonic relationship. The policy trained on the highest-scoring motions achieves 94.6% SR, comparable to the Full Data Baseline but not superior. This suggests that perfect physical scores often correspond to conservative or static motions lacking dynamic richness. Performance peaks significantly at the 60-70% slice with 96.3% SR, where motions strike an optimal balance between physical feasibility and dynamic complexity. As expected, performance plummets in the lowest-scoring slices, with 90-100% dropping to 92.2%. This confirms that physics score effectively identifies toxic data but fails to rank the utility of feasible motions, validating our three-stage design where Stage I functions as a binary filter while Stage II and Stage III identify high-value motions.

### 4.6. Weight Calibration

Assigning weights to different physical error terms is typically heuristic, but not all physical violations are equally detrimental to policy learning. To determine optimal weights, we conducted a sensitivity analysis by filtering

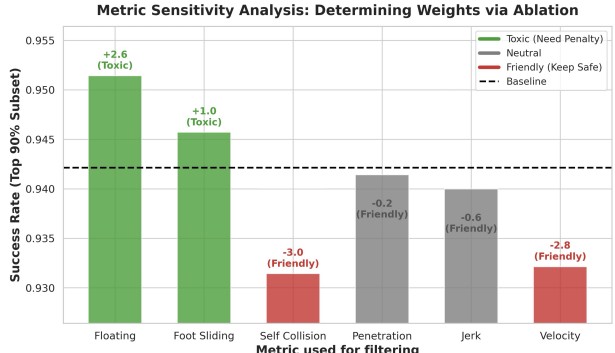

*Figure 6.* **Metric Sensitivity Analysis.** Improvements indicate harmful artifacts that should be penalized heavily, while degradations indicate valuable dynamic motions that should be preserved.

*Table 3.* **Calibrated Weights for Physical Score.** Toxic metrics are penalized heavily while Friendly metrics are penalized lightly to preserve high-energy samples.

| Metric | Impact | Role | Value | Weight ($w_i$) |
|---|---|---|---|---|
| Floating | +2.6 | Toxic | 0.4133 | 24.19 |
| Foot Slide | +1.0 | Toxic | 5.8716 | 1.70 |
| Penetration | -0.2 | Neutral | 0.050 | 216.62 |
| Jerk | -0.6 | Neutral | 3.6025 | 0.28 |
| Velocity | -2.8 | Friendly | 0.0226 | 44.22 |
| Self Collision | -3.0 | Friendly | 5.8343 | 0.17 |

the dataset to keep only the top 90% of motions based on each metric independently, then training policies.

Figure 6 categorizes metrics into three roles based on performance shift. Filtering Floating and Foot Sliding motions significantly improves performance, confirming these artifacts are harmful noise warranting high weights. Surprisingly, filtering high-Velocity or high-Jerk motions degrades performance, implying that violent motions are actually valuable for learning robust dynamics and warrant low weights. Metrics like Penetration and Collision show negligible impact, receiving medium weights as safety constraints. Table 3 presents the calibrated weights derived from this analysis, ensuring that the physical score penalizes artifacts without suppressing useful dynamic behaviors.

## 5. Conclusion

We present **LIMMT**, a data-centric framework for humanoid motion tracking. Our three-stage GQS pipeline filters infeasible motions, embeds them in a semantic space, and selects a compact subset via complexity-weighted sampling. Training on just 3% of curated data outperforms full-corpus baselines. The gains are plug-and-play across trackers and datasets. Motion data is valuable when it is physically feasible, behaviorally diverse, and dynamically rich—not when it merely grows in volume.

# Impact Statement

This paper presents work whose goal is to advance the field of Machine Learning. There are many potential societal consequences of our work, none of which we feel must be specifically highlighted here.

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

# A. Implementation Details

## A.1. Domain Randomization

To improve sim-to-real transfer and policy robustness, we apply domain randomization during PPO training. Table 4 summarizes the randomization ranges. Terrain randomization includes varying floor friction and procedurally generated height maps using Perlin noise. External perturbations are applied at random intervals to simulate unexpected disturbances. Physical properties such as joint friction, link mass, and center-of-mass positions are also randomized to encourage the policy to generalize across different robot configurations.

## A.2. Training Hyperparameters

Table 5 lists the PPO hyperparameters used for training all motion tracking policies. We use a large batch of 32,768 parallel environments to ensure stable gradient estimates. The policy and critic networks share a similar architecture with three hidden layers. All experiments use the same hyperparameters to ensure fair comparison across different data curation strategies.

# B. Additional Experiments

## B.1. Global vs. Clustered Selection

A common heuristic in dataset curation is to first cluster the data to ensure structural diversity, then sample within each cluster. We investigate whether such explicit structural constraints are beneficial for motion tracking compared to global optimization.

Both strategies operate on the same pool of physically feasi-

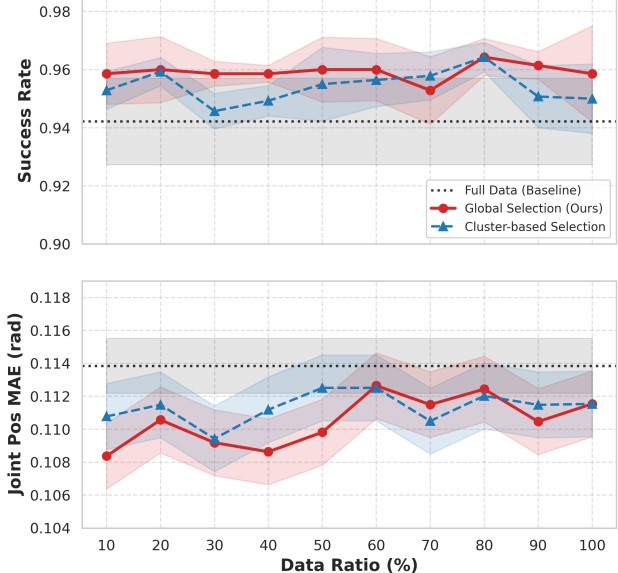

*Figure 7.* **Global vs. Cluster-based Selection.** Global selection outperforms cluster-based selection by naturally re-balancing the distribution rather than inheriting the original dataset's imbalance.

ble motions from Stage I filtering. Cluster-based Selection first clusters motions into 20 groups using K-Means on the embedding space, then applies WFPS within each cluster proportional to cluster size. Global Selection applies WFPS directly on the entire manifold without cluster boundaries.

Figure 7 shows that Global Selection consistently outperforms Cluster-based Selection, particularly in the critical low-data regime. At 10% data, Global WFPS achieves 95.9% SR compared to 95.3% for Cluster-based Selection. We attribute this to the imbalanced nature of motion datasets: easy clusters like locomotion contain massive amounts of data while hard clusters like acrobatics are sparse. By sampling proportionally within clusters, Cluster-based selection

*Table 4.* **Domain Randomizations.**

| Item | Random range |
| --- | --- |
| **Terrains** | |
| Floor friction | $\mathcal{U}(0.3, 2.0)$ |
| Noise scale | $\mathcal{U}(10.0, 16.0)$ |
| Noise octaves | $\mathcal{U}(5.0, 8.0)$ |
| Noise persistence | $\mathcal{U}(0.3, 0.5)$ |
| Noise lacunarity | $\mathcal{U}(2.0, 4.0)$ |
| **External Forces** | |
| Interval range | $\mathcal{U}(5.0, 10.0)$ |
| Velocity magnitude range | $\mathcal{U}(0.1, 1.0)$ |
| **Physical Property Changes** | |
| DoF friction scaling | $\mathcal{U}(0.5, 2.0)$ |
| Armature scaling | $\mathcal{U}(1.0, 1.05)$ |
| Torso CoM position change | $\mathcal{U}(-0.15, 0.15)$ |
| Torso mass change | $\mathcal{U}(-3.0, 6.0)$ |
| Default DoF position jittering | $\mathcal{U}(-0.05, 0.05)$ |

*Table 5.* **Hyperparameter settings for training motion tracker.**

| Hyperparameter | Value |
| --- | --- |
| Env Numbers | 32768 |
| Batch size | 1024 |
| Discount factor $\gamma$ | 0.97 |
| GAE parameter $\lambda$ | 0.95 |
| Clipping parameter $\epsilon$ | 0.2 |
| Policy network size | [1024, 512, 256] |
| Critic network size | [1024, 512, 256] |
| Learning rate | $3 \times 10^{-4}$ |
| Entropy coefficient | 0.01 |
| Weight FPS $\alpha$ | 0.99 |
| Optimizer | Adam |
| Training steps | $2 \times 10^{9}$ |

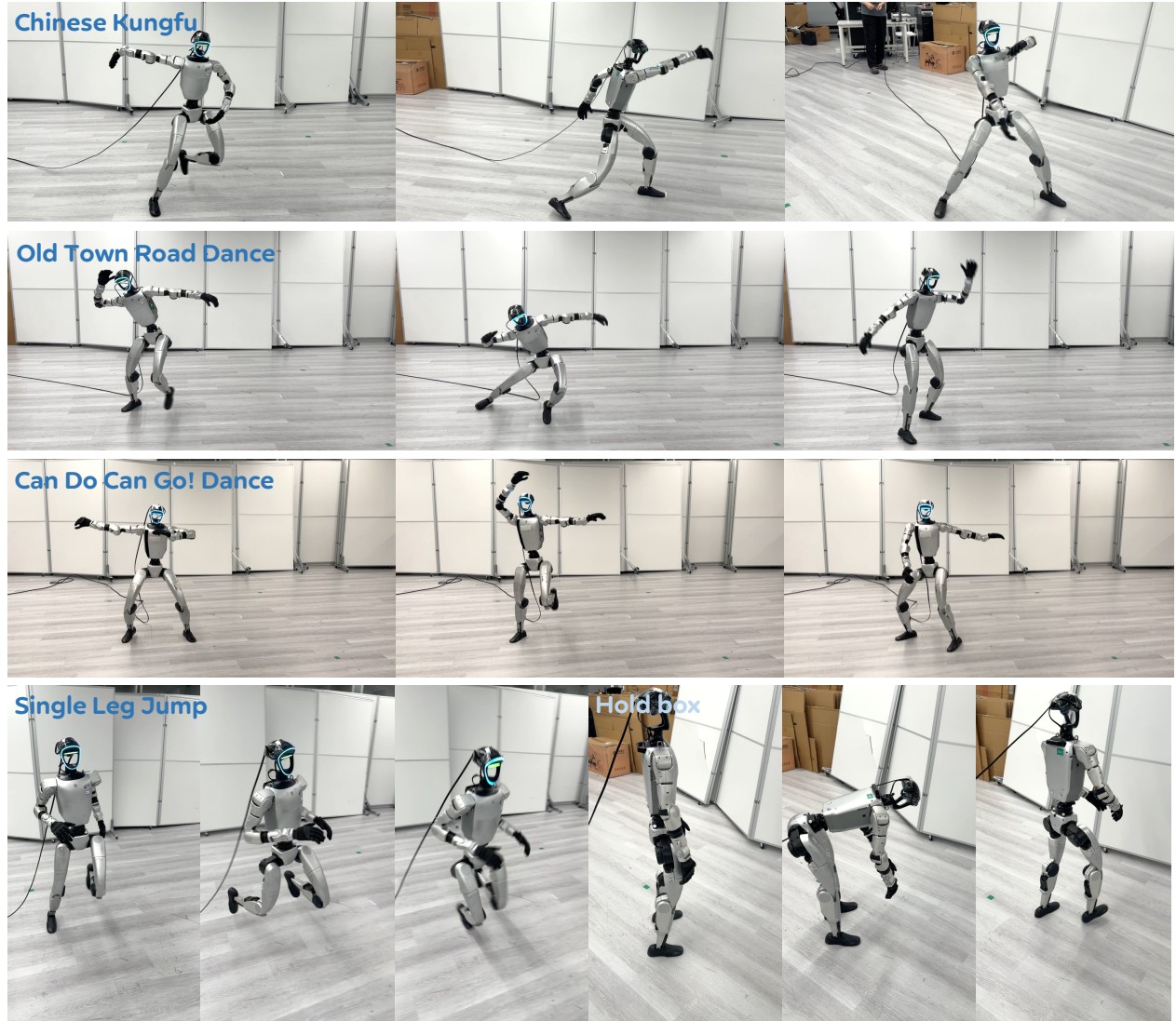

*Figure 8.* **Real-World Motion Tracking.** We deploy the policy trained on GQS-curated data (10%) to the Unitree G1 robot. Each row shows a different motion category: **(a)** Basic motion for daily and teleoperation, **(b)** dancing and expressive movements, **(c)** athletic skills and dynamic motions. The policy successfully transfers to the real robot without fine-tuning, demonstrating that GQS-curated training data not only improves simulation performance but also enhances real-world deployment.

inherits this imbalance. In contrast, Global WFPS acts as an automatic re-balancer, naturally skipping over dense redundant centers and targeting sparse boundaries of the manifold. This finding reinforces that curation should actively flatten the distribution to prioritize information density rather than merely represent the original distribution.

### B.2. Real-World Deployment

To validate the sim-to-real transfer capability of policies trained on GQS-curated data, we deploy our tracker on the physical Unitree G1 humanoid robot. Figure 8 shows qualitative results of real-world motion tracking across diverse motion categories.

The deployed policy demonstrates robust tracking across three motion categories. For **dailylife motion**, the robot accurately reproduces various walking gaits and turning patterns. For **expressive movements**, the policy captures the nuanced dynamics of dancing motions. For **athletic skills**, the robot executes dynamic movements with proper balance and coordination.

Notably, the policy trained on only 10% GQS-curated data achieves successful real-world deployment without any fine-tuning. This zero-shot transfer validates that our data curation strategy not only improves simulation metrics but also produces policies with better generalization to real-world conditions. We attribute this to two factors: (1) physics

*Table 6.* **Real-world tracking on the Unitree G1** (10 trials per category, SR = trials tracked to completion). Policies trained on GQS-curated data (10%) consistently match or surpass Full-Data policies on the physical robot across all four motion categories, while requiring an order of magnitude less data.

| Motion Type | SR (Full) ↑ | SR (GQS 10%) ↑ | MPJPE (Full) ↓ | MPJPE (GQS 10%) ↓ |
|---|---|---|---|---|
| Walking | 10/10 | 10/10 | 0.1037 | **0.0856** |
| Jumping | 8/10 | **9/10** | 0.1453 | **0.1215** |
| Dance 1 | 7/10 | **8/10** | 0.1672 | **0.1384** |
| Dance 2 | 6/10 | **7/10** | 0.1948 | **0.1693** |
| **Average** | 0.7750 | **0.8500** | 0.1528 | **0.1287** |

*Table 7.* **Cross-corpus generalization: train on LaFAN1, test zero-shot on AMASS-test.** LaFAN1 is segmented into 1,532 clips of 576 frames each (matching AMASS' average length). Despite the small training pool ($\approx 10\%$ of AMASS) and substantial game-vs-mocap domain gap, GQS still delivers a consistent improvement.

| Method | Physics Filter | FPS Ratio | Success Rate ↑ | MPJPE (rad) ↓ | MPKPE (mm) ↓ |
|---|---|---|---|---|---|
| LaFAN1 | × | – | 0.8571 | 0.1283 | 42.15 |
| LaFAN1 + **GQS** | ✓ | 70% | **0.8643** | **0.1254** | **41.07** |

filtering removes motions that would cause sim-to-real distribution shift, and (2) complexity-biased selection prioritizes dynamic motions that exercise the full range of robot capabilities.

To complement the qualitative results, we report quantitative real-world tracking metrics on the Unitree G1 across four motion categories (10 trials per category) in Table 6. The 10% GQS-curated policy matches or surpasses the Full-Data policy on every category, with a $+7.5$ SR and $-15.8\%$ MPJPE improvement on average. The advantage is largest on the dynamically demanding Dance categories, where Full-Data policies more often violate balance or contact constraints—consistent with our hypothesis that curated data biases learning toward physically informative trajectories that transfer better to hardware.

### B.3. Cross-Corpus Evaluation on LaFAN1

We additionally test on LaFAN1 (Harvey et al., 2020), a game-oriented mocap corpus that is stylistically distinct from AMASS. Since LaFAN clips are long and contain multiple motion types, we segment them into 576-frame clips (matching AMASS' average length), yielding 1,532 clips—roughly 10% of AMASS' training pool. We then train on LaFAN and evaluate zero-shot on the AMASS test set. Table 7 reports the results: the absolute performance is limited by the small training pool and substantial game-vs-mocap domain gap, but GQS still provides a modest yet consistent improvement, indicating that the data-centric advantage generalizes beyond the AMASS distribution.

### B.4. Hyperparameter Sensitivity

Because GQS is a curation pipeline rather than an end-to-end learned model, its hyperparameters are not just implementation details—they are partly the method itself. We

therefore provide systematic sensitivity analyses for the four most consequential design choices: (a) the physics threshold $S_{\text{phy}}$, (b) the foot-sliding height threshold, (c) the PAE window size, and (d) the HME embedding half-dimension $k$. All variants are evaluated on AMASS; (c) and (d) are trained under a reduced compute budget of $1 \times 10^9$ steps with Any2Track at 10% data, since changing either requires retraining both the PAE and the downstream tracker from scratch.

**Threshold $S_{\text{phy}} = 90$ is not arbitrary.** Each penalty weight is calibrated as $w_i = 10/\text{Boundary}_i$, where $\text{Boundary}_i$ is the maximum acceptable value for a Toxic artifact. With this calibration, any clip with $S_{\text{phy}} < 90$ is guaranteed to contain at least one severe physical violation. Table 8(a) confirms that 90 is near-optimal among $\{80, 85, 90, 95, 99\}$.

**Foot-sliding threshold $5\,\text{cm}$ matches the robot.** The threshold is grounded in the Unitree G1 URDF: the ankle-to-ground clearance ranges from $3.50\,\text{cm}$ to $5.26\,\text{cm}$ depending on joint configuration. We adopt $5\,\text{cm}$ so that only motions with feet *clearly* below the ground plane are penalized. Table 8(b) shows that performance is essentially flat between 2 and $10\,\text{cm}$, with $5\,\text{cm}$ slightly best.

**PAE window and embedding dimension.** Tables 8(c)–(d) show that 4 s windows and $k = 8$ are near-optimal, with smooth degradation outside the chosen settings.

### B.5. Adaptive Ratio Selection (ARS)

A practical question is: when applying GQS to a *new* dataset, how should one pick the subset ratio without expensive trial-and-error training? We present a lightweight, post-hoc heuristic, **Adaptive Ratio Selection (ARS)**, that estimates a reasonable ratio purely from embedding statistics.

*Table 8.* **Hyperparameter sensitivity analyses on AMASS.** Across all four design choices, performance is stable within a reasonable range and our chosen defaults (highlighted) consistently perform at or near the optimum, confirming that the GQS design is robust rather than brittle. The PAE window and embedding dimension panels are trained under a reduced budget of $1 \times 10^9$ steps with Any2Track at 10% data (changing either requires retraining both the PAE and the downstream tracker from scratch).

(a) Physics threshold $S_{\text{phy}}$.

| $S_{\text{phy}}$ | SR ↑ | MPJPE ↓ | MPKPE ↓ |
|---|---|---|---|
| 80 | 0.8107 | 0.1189 | 51.62 |
| 85 | 0.8083 | 0.1185 | 52.02 |
| **90 (ours)** | **0.8250** | **0.1165** | **48.63** |
| 95 | 0.8095 | 0.1193 | 52.49 |
| 99 | 0.8178 | 0.1164 | 49.09 |

(b) Foot-sliding height threshold.

| Threshold | SR ↑ | MPJPE ↓ | MPKPE ↓ |
|---|---|---|---|
| 0 cm | 0.8131 | 0.1182 | 50.34 |
| 2 cm | 0.8202 | 0.1170 | 49.21 |
| **5 cm (ours)** | **0.8250** | **0.1165** | **48.63** |
| 10 cm | 0.8190 | 0.1173 | 49.56 |

(c) PAE window size.

| Window | SR ↑ | MPJPE ↓ | MPKPE ↓ |
|---|---|---|---|
| 1 s | 0.8340 | 0.1077 | 41.42 |
| 2 s | 0.8702 | 0.1029 | 36.82 |
| **4 s (ours)** | **0.8881** | **0.0968** | **33.50** |
| 8 s | 0.8393 | 0.1099 | 45.59 |

(d) HME embedding half-dimension $k$ ($D = 2k$).

| $k$ | SR ↑ | MPJPE ↓ | MPKPE ↓ |
|---|---|---|---|
| 4 | 0.8607 | 0.1011 | 37.52 |
| **8 (ours)** | **0.8881** | **0.0968** | **33.50** |
| 16 | 0.8797 | 0.1006 | 35.32 |
| 32 | 0.8809 | 0.0967 | 34.17 |

*Table 9.* **Adaptive Ratio Selection (ARS) predictions.** $d_{\text{eff}}$ is the number of PCA components explaining 95% of variance in the HME embedding, and $D = 2k$ is the embedding dimension. EDR $= d_{\text{eff}}/D$ measures how much of the embedding capacity is occupied by genuine motion diversity. The ARS-predicted ratio $r^\star = c \cdot \text{EDR}^2$ (with $c \approx 0.5$) matches the empirically optimal subset ratio observed in our experiments on both datasets.

| Dataset | $N$ | $d_{\text{eff}} / D$ | EDR | $r^\star$ (predicted) | Empirical optimum |
|---|---|---|---|---|---|
| AMASS (Mahmood et al., 2019) | 24,197 | 15 / 32 | 0.47 | 11.0% | 10% |
| PHUMA (Lee et al., 2025) | 75,886 | 26 / 32 | 0.81 | 32.8% | 30% |

**Core insight.** The optimal ratio depends on the *data homogeneity* of the corpus after physics filtering. A dataset with high internal redundancy (many similar walking/standing clips with a few complex ones) benefits from aggressive filtering, while a well-curated, diverse dataset needs a larger fraction to preserve coverage.

**Effective Diversity Ratio (EDR).** We operationalize homogeneity through the EDR,

$$\text{EDR} = \frac{d_{\text{eff}}}{D}, \qquad (6)$$

where $d_{\text{eff}}$ is the number of PCA components explaining 95% of the variance of the HME embeddings, and $D = 2k$ is the embedding dimension. A low EDR indicates concentrated, redundant data; a high EDR indicates broadly diverse data. The predicted optimal ratio is then

$$r^\star = c \cdot \text{EDR}^2, \qquad c \approx 0.5. \qquad (7)$$

**Empirical validation.** Table 9 reports EDR and the ARS prediction on AMASS and PHUMA. AMASS uses only 15 of 32 embedding dimensions (EDR $= 0.47$, complexity coefficient of variation CV $= 1.19$), indicating high redundancy; ARS predicts $r^\star = 11.0\%$, closely matching the empirically optimal 10%. PHUMA, being curated and physics-grounded, occupies 26/32 dimensions

(EDR $= 0.81$, CV $= 0.70$); ARS predicts $r^\star = 32.8\%$, matching the empirically optimal 30%.

**Practical procedure.** For a new dataset, the recipe is: *(1)* run GQS Stage I and Stage II; *(2)* compute PCA on the embeddings to obtain $d_{\text{eff}}$ (95% variance); *(3)* apply $r^\star = 0.5 \cdot (d_{\text{eff}}/D)^2$; *(4)* use $r^\star$ as the Stage III selection ratio. ARS adds a single PCA on top of the existing pipeline and removes the need for ratio sweeps. We caveat that this heuristic is validated on two corpora; broader empirical study is warranted.

## C. Future Work

Our quality assessment is currently rule-based. A promising direction is to collect human preference data and train a preference-driven reward model, analogous to RLHF (Christiano et al., 2017; Xiong et al., 2024), enabling finer-grained quality evaluation and more informative learning signals. Another valuable avenue is to repair low-quality motions, or to pretrain on large-scale low-quality corpora to distill the latent dark knowledge they contain. Furthermore, this concept can be applied to Vision-Language-Action models in robotic arm manipulation (Zhang et al., 2025a; Qi et al., 2025; Sun et al., 2026; Zhang et al., 2026a; Qi et al., 2024). Regardless of the field, data quality should always be treated as a critical component.

