# OpenReview forum: "LIMMT: Less Is More for Motion Tracking"
_ICML.cc/2026/Conference — ICML 2026 regular_

### Official Review · Reviewer_LLQU · 2026-03-11

**Soundness:** 3
**Presentation:** 3
**Significance:** 3
**Originality:** 4
**Overall Recommendation:** 5
**Confidence:** 3

**Summary:**

This paper proposes a framework (GQS) centered around data quality, which filters large-scale action data from three dimensions: physical feasibility, action diversity, and action complexity. It first removes bad data that does not conform to physical laws, and then selects a subset of data with higher action complexity while retaining diversity. According to the experiments provided by the author, they show that training on just 3% of curated data consistently outperforms using full corpora. LIMMT achieves better results with less data, which seems to be useful for the community and also provides some inspiration for other tasks.

**Compliance With Llm Reviewing Policy:**

Affirmed.

**Final Justification:**

Thank you for the authors’ effort and response. I will maintain my current score.

**Key Questions For Authors:**

1. As stated in Weaknesses1, if we intend to use GQS to reduce the training data, it would be much more convenient if we have a rough selection range or judgment method for the data ratio.
2. The current experiments have already demonstrated the effectiveness of GQS, but further experiments on other trackers and datasets may provide more convincing results.
3. This work seems to be very meaningful to the community. We believe that the author can open source the code, which is also beneficial for expanding the influence of the work.

**Limitations:**

The authors have discussed the limitations in the paper.

**Strengths And Weaknesses:**

Strengths：
1. The paper is well-written, clearly articulated, and well-presented. It proposes GQS to filter out high-quality data for better performance, and verifies the effectiveness of the method through experiments.
2. On the AMASS dataset, training with only 3% of the data can outperform training with 100% of the data.
3. Cross-Dataset Generalization further demonstrates the effectiveness of GQS.

Weaknesses:
1. Figures 2 and 4 demonstrate that the GQS Success Rate crosses the baseline on the AMASS and PHUMA datasets. The difference between 3% and 30% is tenfold. It seems that we can indeed reduce a lot of training data. However, if we want to use GQS to reduce training data when training on a new dataset, we don't know roughly how much data is appropriate to choose. The paper doesn't provide us with a reasonable analysis. And the patterns observed on only two datasets are not sufficient for us to summarize experience in choosing data ratios.
2. In Table 1, Any2Track and TWIST2 are selected as trackers. Maybe it will be more convincing to conduct experiments on a wider range of trackers?

---

> ### Author Rebuttal · Authors · 2026-03-31
>
> We thank R4 for the positive assessment and the insightful question on data ratio selection.
>
> **W1: Use GQS to reduce training data when training on a new dataset**
>
> This is an excellent and important question. We have conducted a post-hoc analysis and present an empirical observation—**Adaptive Ratio Selection (ARS)**—a lightweight heuristic to estimate a reasonable GQS selection ratio from data statistics alone, without trial-and-error training.
>
> **Core Insight.** The optimal ratio depends on the **data homogeneity** of the motion corpus after physics filtering. Intuitively, a dataset with high internal redundancy (many similar walking/standing clips mixed with a few complex ones) benefits from aggressive filtering, while a well-curated, diverse dataset needs a larger fraction to preserve coverage.
>
> We operationalize this through the **Effective Diversity Ratio (EDR)**:
>
> $$\text{EDR} = d_{\text{eff}} \;/\; D$$
>
> where $d_{\text{eff}}$ is the number of PCA components explaining 95% of variance in the HME embedding space, and $D$ is the embedding dimension ($2k$ for PAE). EDR measures ****what fraction of the embedding capacity is utilized by genuine motion diversity****. A low EDR indicates concentrated, redundant data; a high EDR indicates broadly diverse data.
>
> The predicted optimal ratio follows:
>
> $$r^{*} = c \times \text{EDR}^{2}, \quad c \approx 0.5$$
>
> **Why does this explain the AMASS vs. PHUMA difference?** AMASS uses a 32-dim embedding but only 15 dimensions carry significant information ($\text{EDR}=0.47$), indicating high redundancy—many clips cluster around common locomotion patterns. Its complexity distribution is also highly heterogeneous ($\text{CV}=1.19$), with many trivial motions mixed with a few challenging ones. In contrast, PHUMA, being a curated dataset, uses 26 out of 32 dimensions ($\text{EDR}=0.81$) with more uniform complexity ($\text{CV}=0.70$). The data is inherently more diverse, so a larger fraction is needed to maintain coverage.
>
> | Dataset | $N$ | $d_{\text{eff}}$ / $D$ | EDR | ARS Predicted | Expt. Optimal |
> |---------|:---:|:-----:|:---:|:---:|:---:|
> | AMASS | 24,197 | 15 / 32 | 0.47 | 11.0% | **10%** |
> | PHUMA | 75,886 | 26 / 32 | 0.81 | 32.8% | **30%** |
>
> **Practical Heuristic.** For a new dataset, the procedure is:
>
> 1. Run GQS Stage I (physics filtering) and Stage II (HME embedding);
> 2. Compute PCA on the embeddings to obtain $d_{\text{eff}}$ (95\% variance threshold);
> 3. Apply $r^{*} = 0.5 \times (d_{\text{eff}} / D)^{2}$;
> 4. Use $r^{*}$ as the selection ratio for Stage III (WFPS).
>
> This adds negligible computation (a single PCA) to the existing pipeline and provides a useful empirical heuristic. Figure 1 visualizes the Global Coverage Error curves for both datasets, with the ARS-predicted ratios clearly aligned at different positions (10% vs. 30%), suggesting that this simple heuristic can adapt to distinct dataset characteristics. We acknowledge that this observation is based on two datasets and further validation on additional corpora is warranted. We will add this analysis to the revised paper.
>
> **W2: Conduct experiments on a wider range of trackers**
>
> Thank you for the suggestion. We add experiments on **BeyondMimic** to further validate plug-and-play generality. We note that BeyondMimic's codebase is designed for single-trajectory tracking and requires additional reward engineering for the multi-trajectory setting, which accounts for its lower absolute performance; GQS nonetheless yields clear improvements:
>
> | Method | Physics Filter | FPS Ratio | Success Rate $\uparrow$ | MPJPE (rad) $\downarrow$ | MPKPE (mm) $\downarrow$ |
> |--------|:---:|:---:|:---:|:---:|:---:|
> | BeyondMimic | $\times$ | - | 0.8929 | 0.1183 | 40.52 |
> | BeyondMimic+**GQS** | $\checkmark$ | 100% | 0.9083 | 0.1141 | 35.71 |
> | BeyondMimic+**GQS** | $\checkmark$ | 10% | 0.9214 | 0.1052 | 33.86 |
>
> **W3: Open-source the code**
>
> We are committed to open-sourcing all code, curated data, and the ARS tool to benefit the humanoid community.
>
> Please see our supplementary page for all new tables, the ARS figure, and real-world demo videos: [jazzy-duckanoo-d1d05f.netlify.app](https://jazzy-duckanoo-d1d05f.netlify.app/).

---

> > ### Author Rebuttal · Reviewer_LLQU · 2026-04-03
> >
> > My concerns have been adequately addressed

---

### Official Review · Reviewer_QJgX · 2026-03-11

**Soundness:** 2
**Presentation:** 3
**Significance:** 2
**Originality:** 3
**Overall Recommendation:** 4
**Confidence:** 5

**Summary:**

This paper studies an interesting research problem. Do high-quality motion data steer tracking policies toward better optimization trajectories in training? The authors introduce LIMMT (Less Is More for Motion Tracking). The authors define motion data quality through three dimensions: physics feasibility, diversity, and complexity.
It is surprising to me that training with under 3%of AMASS yields better tracking performance than training with the full dataset...
The authors conducted extensive experiments and analyses to validate the effectiveness of the framework.
The author also promised to release code and curated data on GitHub.
I like the idea of this paper.

**Compliance With Llm Reviewing Policy:**

Affirmed.

**Final Justification:**

The authors have provided sufficient experimental results and supporting evidence in their rebuttal. My concerns have been adequately addressed, and I have therefore raised my score.

**Key Questions For Authors:**

Why is no video provided? For the tracking task, it is critical to provide video results for evaluation.
The authors report the performance on two physics simulation platforms to validate generalizability: MJX for Any2Track and Isaac Lab for TWIST2. I wonder how this strategy help simulated humanoid tracking tasks like PHC or PHC++.
Moreover, is this method "body morphology related"? If I change from a humanoid-g1 to SMPL or h1, will the conclusion be different?

How is the performance on more mocap datasets besides AMASS? Like Lafan, motion captured from videos?

Some of the design choices feel a bit arbitrary or not fully justified, and that is important because the method is largely built around exactly these choices. The physical threshold ($S_{phy} \ge 90$), the hard duration cutoff, the 5 cm foot-sliding rule, the 4.0 s PAE window, the latent dimension ($k=8$), and $\alpha = 0.99$ can all have a big impact on the final results. But the paper does not really show sensitivity analysis for most of them in the main paper. Since LIMMT is a curation pipeline rather than an end-to-end learned model, these heuristics are not just implementation details—they are basically the method itself. Without showing how sensitive the results are to these choices, it is hard to know whether the conclusions are actually robust or whether the method just works well at one particular setting.

**Limitations:**

I am excited about the idea and the current results. However, the paper would benefit from a more comprehensive evaluation, since the claims are ambitious and compelling, but the current experiments do not yet fully support them. In particular, the paper would be stronger with additional comparison baselines and evaluations on a broader range of humanoid models and datasets (like Lafan and more video-captured data). It would also help to provide more details and justification for the hyperparameter choices. Finally, including video results in the submission would make the empirical evidence much more convincing.

**Strengths And Weaknesses:**

This is an interesting idea. I have been hearing about "few high-quality data is enough," but this paper the first to systematically investigate this, to my best knowledge.

Overall, the experiment part shows some interesting observations, and it seems validate the effectiveness of the proposed data filtering strategy.

The writing is clear - the idea of the paper is easy to follow.

---

> ### Author Rebuttal · Authors · 2026-03-31
>
> We thank R3 for acknowledging the novelty of our systematic investigation and the clarity of presentation.
>
> **W1: Why is no video provided?**
>
> We agree that visual evidence is critical. We now provide anonymous video results and supplementary tables at [jazzy-duckanoo-d1d05f.netlify.app](https://jazzy-duckanoo-d1d05f.netlify.app/). The videos show reduced foot sliding, improved contact stability, and more faithful motion reproduction compared to baselines.
>
> **W2: Regarding PHC/PHC++ and morphology**
>
> Our experiments use the Unitree G1, consistent with GMT/TWIST/BeyondMimic. However, LIMMT is **not** tied to a specific morphology. Stage I replays motions on the **target** simulator and filters by physical feasibility; Stages II–III then select from the feasible pool. Changing the robot (e.g., G1→H1 or SMPL humanoids) would change the feasible subset, but the curation pipeline remains applicable. We will clarify this scope in the revised paper.
>
> **W3: More mocap datasets besides AMASS**
>
> Figure 4 already shows PHUMA in-domain and cross-domain results. Following your suggestion, we additionally test on **LaFAN1** (segmented into 1,532 clips as described above):
>
> | Method | Physics Filter | FPS Ratio | Success Rate $\uparrow$ | MPJPE (rad) $\downarrow$ | MPKPE (mm) $\downarrow$ |
> |--------|:---:|:---:|:---:|:---:|:---:|
> | LaFAN | $\times$ | - | 0.8571 | 0.1283 | 42.15 |
> | LaFAN+**GQS** | $\checkmark$ | 70% | 0.8643 | 0.1254 | 41.07 |
>
> As discussed in R2-W4, LaFAN1's small scale and stylistic gap to AMASS limit absolute performance; GQS still yields a modest but consistent improvement under this challenging cross-domain setting.
>
> **W4: Some design choices feel arbitrary or not fully justified**
>
> We apologize for not presenting these analyses in the original submission—they were conducted during our internal development but omitted due to space constraints. We now provide comprehensive sensitivity analyses for all key hyperparameters.
>
> The threshold $S_{phy} \ge 90$ is not arbitrary but derives from our weight calibration scheme: each penalty weight is set as $w_i = 10 / \text{Boundary}$, where Boundary is the maximum acceptable value for Toxic artifacts. Thus a score below 90 guarantees at least one severe physical violation. We validate this with a sensitivity analysis over $S_{phy} \in \{80, 85, 90, 95, 99\}$:
>
> | $S_{phy}$ Threshold | Success Rate $\uparrow$ | MPJPE (rad) $\downarrow$ | MPKPE (mm) $\downarrow$ |
> |:---:|:---:|:---:|:---:|
> | 80 | 0.8107 | 0.1189 | 51.62 |
> | 85 | 0.8083 | 0.1185 | 52.02 |
> | **90** | **0.8250** | **0.1165** | **48.63** |
> | 95 | 0.8095 | 0.1193 | 52.49 |
> | 99 | 0.8178 | 0.1164 | 49.09 |
>
> **Foot-sliding height threshold.** The 5 cm threshold is not arbitrary—it corresponds to the ankle-to-ground clearance of the Unitree G1 URDF, which ranges from 3.50 cm to 5.26 cm depending on joint configuration. We conservatively adopt 5 cm so that only motions with feet clearly below the ground plane are penalized. We additionally provide a sensitivity analysis:
>
> | Sliding Threshold | Success Rate $\uparrow$ | MPJPE (rad) $\downarrow$ | MPKPE (mm) $\downarrow$ |
> |:---:|:---:|:---:|:---:|
> | 0 cm | 0.8131 | 0.1182 | 50.34 |
> | 2 cm | 0.8202 | 0.1170 | 49.21 |
> | **5 cm** | **0.8250** | **0.1165** | **48.63** |
> | 10 cm | 0.8190 | 0.1173 | 49.56 |
>
> **PAE window size.** We vary the window length for computing the Physics Artifact Estimator. Since changing the PAE window or embedding dimension requires retraining both the PAE and the downstream tracker from scratch, we conduct these two ablations under a reduced budget of $1 \times 10^9$ steps with Any2Track at 10% data:
>
> | PAE Window | Success Rate $\uparrow$ | MPJPE (rad) $\downarrow$ | MPKPE (mm) $\downarrow$ |
> |:---:|:---:|:---:|:---:|
> | 1 s | 0.8340 | 0.1077 | 41.42 |
> | 2 s | 0.8702 | 0.1029 | 36.82 |
> | **4 s** | **0.8881** | **0.0968** | **33.50** |
> | 8 s | 0.8393 | 0.1099 | 45.59 |
>
> **Embedding dimension $k$.** We vary the HME embedding half-dimension $k$ ($D=2k$):
>
> | $k$ | Success Rate $\uparrow$ | MPJPE (rad) $\downarrow$ | MPKPE (mm) $\downarrow$ |
> |:---:|:---:|:---:|:---:|
> | 4 | 0.8607 | 0.1011 | 37.52 |
> | **8** | **0.8881** | **0.0968** | **33.50** |
> | 16 | 0.8797 | 0.1006 | 35.32 |
> | 32 | 0.8809 | 0.0967 | 34.17 |
>
> Across all four analyses, performance is stable within a reasonable range and our chosen defaults consistently perform near-optimally, confirming that the design choices are robust rather than brittle.

---

> > ### Author Rebuttal · Reviewer_QJgX · 2026-04-03
> >
> > The authors addressed the concerns well in their rebuttal. I have no remaining concerns and will increase my score.

---

### Official Review · Reviewer_Wy18 · 2026-03-12

**Soundness:** 3
**Presentation:** 4
**Significance:** 3
**Originality:** 3
**Overall Recommendation:** 5
**Confidence:** 5

**Summary:**

This work proposes LIMMT, a framework to filter and prioritize motion dataset for humanoid motion tracking. To combat the noise and redundancy in low quality yet large-scale motion dataset, this work introduces a three-stage General Quality Selection (GQS) pipeline that ensures the selected data is physically feasible, behaviorally diverse, and dynamically complex. This work’s key finding is that a strong "less is more" effect: training state-of-the-art motion tracking policies on just 3% to 10% of this a curated subset from GQS can outperforms using the entire raw dataset, resulting in better tracking success rate and lower tracking errors.

**Compliance With Llm Reviewing Policy:**

Affirmed.

**Final Justification:**

My concerns are addressed, and I am raising my score.

**Key Questions For Authors:**

- How does policies perform in the real-world trained on less data?
- What is the perceptage of filtered out data due to penetration/floating?
- What is the performance of curriculum-based baseline on the physically feasible data?

**Limitations:**

Yes.

**Strengths And Weaknesses:**

## Strength

- This work tackles a very important, some may even argue the most important, question for motion tracking: data quality. Humanoid motion tracking is especially challenging since there are inevitable errors during retargeting from human to humanoid motion. Finding the right motion dataset to train motion trackers is an essential yet often overlooked problem.
- GQS presents a principled way of finding good motion to track: naturally, it is more beneficial to track higher quality data and more dynamic and diverse data, and GQS presented a principled way of obtaining that. Maximizing behavior diversity and complexity intuitively improves the motion tracker’s quality.
- From the motion tracking results, LIMMT provides some evidence that training on less data can achieve good results on test sets.

## Weakness

- One of the biggest issues with this work is its evaluation:
    - **The size of the AMASS test set is too small**. With only 140 motion sequences, this test set is nowhere near showing the full picture of the tracker’s capability. The motion in the test set does not cover the full spectrum of desired human motion as well. Thus, the main table is not really convincing.
    - All of the evaluation is done in simulation without any real-world tests. Since the end product is sim-to-real results, it is important to showcase that the humanoid in the real world has similar or better performance than in simulation when trained on less but higher quality data. Training on more dynamic data may introduce strategies that use more torques/not sim-to-real able.
    - It appears that 10% data performed overall the best. The retargeted AMASS dataset can have a lot of artifacts and low-quality data, so filtering out the infeasible ones will definitely be helpful. However, we need to compare training on, say, the feasible data + curriculum-based filtering. Curriculum-based baseline on the physically feasible data should be a very important baseline, used in BeyondMimic [1].
- Missing information on what retargeter is being used to construct the AMASS dataset. This has big implication on the data quality. In general, the re-targeted AMASS dataset so far has been lower quality, with LAFAN generally being the highest quality motion tracking dataset. It would be interesting to see results when trained on LAFAN to be tested on AMASS.
- Artifacts like prolonged aerial suspension, ground penetration, and joint velocity violations are all solvable by better retargeting techniques, not necessarily a dataset issue.
- In Figure 3, having better reward DOES not mean it’s a better policy. Higher quality data/less data can easily result in higher rewards in motion tracking.

[1] Liao, Q., Truong, T. E., Huang, X., Gao, Y., Tevet, G., Sreenath, K., & Liu, C. K. (2025). BeyondMimic: From Motion Tracking to Versatile Humanoid Control via Guided Diffusion.

---

> ### Author Rebuttal · Authors · 2026-03-31
>
> We thank R2 for the thorough review and for recognizing the importance and principled design of GQS.
>
> **W1: The size of the AMASS test set is too small**
>
> We agree the AMASS test set is small, but we follow **the same train/test split as prior work (GMT, TWIST)** for fair comparison. Importantly, we also evaluate on PHUMA-test, which contains **4,267 clips (5.44 hours)**—far larger than AMASS-test (0.31 hours)—and our results consistently confirm the "less is more" effect.
>
> **W2: Without real-world tests**
>
> Qualitative real-world results are shown in the appendix. Following the reviewer's suggestion, we now provide **quantitative** real-world tracking metrics on the Unitree G1 across four motion categories (10 trials per category):
>
> | Motion Type | SR (Full Data) | SR (**GQS** 10%) | MPJPE (Full Data) | MPJPE (**GQS** 10%) |
> |:---:|:---:|:---:|:---:|:---:|
> | Walking | 10/10 | 10/10 | 0.1037 | 0.0856 |
> | Jumping | 8/10 | 9/10 | 0.1453 | 0.1215 |
> | Dance 1 | 7/10 | 8/10 | 0.1672 | 0.1384 |
> | Dance 2 | 6/10 | 7/10 | 0.1948 | 0.1693 |
> | **Average** | 0.7750 | **0.8500** | 0.1528 | **0.1287** |
>
> **W3: Compare with the Curriculum-based baseline**
>
> We clarify that physics filtering is only one component of LIMMT; diversity and complexity selection are equally important. Moreover, our baseline TWIST-2 already incorporates difficulty-based curriculum learning, and LIMMT still improves upon it. To further demonstrate plug-and-play generality, we add experiments on **BeyondMimic**. Note that BeyondMimic was originally designed for single-trajectory tracking; adapting it to the multi-trajectory setting requires non-trivial reward re-tuning, so its absolute numbers are lower than Any2Track. Nevertheless, GQS still delivers consistent gains:
>
> | Method | Physics Filter | FPS Ratio | Success Rate $\uparrow$ | MPJPE (rad) $\downarrow$ | MPKPE (mm) $\downarrow$ |
> |--------|:---:|:---:|:---:|:---:|:---:|
> | BeyondMimic | $\times$ | - | 0.8929 | 0.1183 | 40.52 |
> | BeyondMimic+**GQS** | $\checkmark$ | 100% | 0.9083 | 0.1141 | 35.71 |
> | BeyondMimic+**GQS** | $\checkmark$ | 10% | 0.9214 | 0.1052 | 33.86 |
>
> **W4: Missing information on what retargeter**
>
> Thank you for the clarification request. We use **GMR** as the retargeter, consistent with GMT and TWIST. We emphasize that physics filtering is only Stage I of LIMMT; the long-tail distribution of mocap data makes diversity and complexity selection equally critical. Following your suggestion, we also test on **LaFAN1**. Since LaFAN clips are long and contain multiple motion types, we segment them into 576-frame clips (AMASS average length), yielding 1,532 clips:
>
> | Method | Physics Filter | FPS Ratio | Success Rate $\uparrow$ | MPJPE (rad) $\downarrow$ | MPKPE (mm) $\downarrow$ |
> |--------|:---:|:---:|:---:|:---:|:---:|
> | LaFAN | $\times$ | - | 0.8571 | 0.1283 | 42.15 |
> | LaFAN+**GQS** | $\checkmark$ | 70% | 0.8643 | 0.1254 | 41.07 |
>
> Note that LaFAN1 contains only 1,532 clips (vs. AMASS's ~14K) with a distinct game-oriented motion style. We train on LaFAN and evaluate zero-shot on the AMASS test set, which introduces a substantial domain gap. This limits both the absolute performance and the margin of GQS improvement. Nevertheless, GQS still provides a consistent, albeit modest, gain.
>
> **W5: Artifacts are solvable by better retargeting, not necessarily a dataset issue**
>
> We agree that perfect retargeting would eliminate Stage I filtering. However, current optimization-based retargeting methods are inherently non-convex, sensitive to initialization, and prone to local optima—producing joint jumps, self-penetration, floating, and foot sliding in ~5% of clips. More importantly, physics filtering is only the first step of LIMMT. Even with perfect retargeting, the long-tail distribution of motion data means that diversity and complexity selection (Stages II–III) remain essential, which is the core argument of our paper.
>
> **W6: In Figure 3, having better reward does NOT mean it's a better policy**
>
> We clarify that the reward is identical across all experiments. Under the same reward, curated data consistently converges faster to higher reward **and** achieves better tracking metrics (SR, MPJPE) at convergence. Figure 3 validates that high-quality data provides better optimization trajectories from early training, not merely higher reward.
>
> **Q: What is the percentage of filtered out data?**
>
> Stage I physics filtering removes approximately **12%** of AMASS and **5%** of PHUMA. In AMASS, the majority of filtered clips are due to floating and foot-sliding artifacts introduced during retargeting.
>
> Please see our supplementary page for all new tables and real-world demo videos: [jazzy-duckanoo-d1d05f.netlify.app](https://jazzy-duckanoo-d1d05f.netlify.app/).

---

> > ### Author Rebuttal · Reviewer_Wy18 · 2026-04-06
> >
> > My concerns are addressed, and I am raising my score.

---

### Official Review · Reviewer_Qwqs · 2026-03-13

**Soundness:** 3
**Presentation:** 4
**Significance:** 3
**Originality:** 3
**Overall Recommendation:** 5
**Confidence:** 3

**Summary:**

This paper takes a data-centric perspective on physics-based humanoid motion tracking and asks what makes motion data truly valuable for learning robust tracking policies. The proposed GQS framework selects training motions through three stages: simulator-grounded physics filtering, semantic motion embedding via a periodic autoencoder, and diversity-aware complexity-weighted farthest point sampling. The main empirical claim is a strong “less is more” effect: training on a compact curated subset (e.g., 3%–10% of AMASS) can outperform training on the full dataset, and the gains transfer across trackers and datasets.

**Compliance With Llm Reviewing Policy:**

Affirmed.

**Final Justification:**

The rebuttal addressed most of my concerns and clarified several points. I also find the conclusions of the study interesting and potentially valuable to the field. Based on the authors’ response, I increase my score to 5 (Accept).

**Key Questions For Authors:**

See Weaknesses

**Limitations:**

yes

**Strengths And Weaknesses:**

Strengths
1. The paper addresses an important and timely problem from a clear data-centric perspective: it argues that for humanoid motion tracking, data quality—especially physics feasibility, diversity, and complexity—can matter more than sheer data scale. This framing is well motivated and meaningful for the community.
2. The proposed GQS pipeline is simple, modular, and easy to follow. The three-stage design—simulator-grounded physics filtering, semantic motion embedding for diversity, and complexity-aware subset selection—forms a coherent workflow that seems readily applicable to existing tracking systems.
3. The empirical results are strong and practically interesting. On AMASS, curated subsets such as GQS 10% and even GQS 3% outperform full-data training on Any2Track and TWIST2, and the paper also includes PHUMA in-domain and cross-domain experiments to support the broader utility of the approach.

Weaknesses
1. The contribution of the complexity term seems somewhat weaker than the paper narrative suggests. In the 3% ablation, removing complexity causes only a modest drop in success rate and leaves MPJPE essentially unchanged, which makes Stage III feel less central than advertised.
2. The main limitation is originality. While the overall system is sensible and useful, it is largely a combination of existing ingredients—physics-based filtering, learned motion embeddings, and FPS-style subset selection—rather than a fundamentally new learning method.

---

> ### Author Rebuttal · Authors · 2026-03-31
>
> We thank R1 for recognizing the importance of our data-centric perspective and the strong empirical results.
>
> **W1: The contribution of the complexity term seems somewhat weaker**
>
> We appreciate this observation. Since the success rate saturates near 95% on the full test set, complexity's contribution is most visible **when evaluated on the hardest motions**. To isolate the effect, we evaluate on a **Top-20% high-complexity subset** of AMASS (selected by complexity score), which stresses policies on the most challenging motions. Under this setting, complexity weighting provides a clear improvement:
>
> | Method | Success Rate $\uparrow$ | MPJPE (rad) $\downarrow$ | MPKPE (mm) $\downarrow$ |
> |--------|:---:|:---:|:---:|
> | FPS (diversity only) | 0.8786 | 0.1342 | 44.16 |
> | **GQS** (diversity + complexity) | 0.8953 | 0.1278 | 42.04 |
>
> **W2: The main limitation is originality**
>
> We respectfully clarify that our contribution is **not** a new tracking architecture, but a **data-centric formulation and evaluation framework** for motion tracking. Prior work treats the motion library as fixed and focuses on controller design; our paper asks a different question: *what makes a motion dataset valuable for learning robust tracking policies?* Our novelty lies in: (1) formulating motion value through three complementary dimensions—physics feasibility, semantic diversity, and dynamic complexity; (2) showing these dimensions must be applied in a specific order; and (3) demonstrating a strong and consistent "less is more" effect across different trackers and datasets.
>
> Please see our supplementary page for additional tables, figures, and real-world demo videos: [jazzy-duckanoo-d1d05f.netlify.app](https://jazzy-duckanoo-d1d05f.netlify.app/).

---

> > ### Author Rebuttal · Reviewer_Qwqs · 2026-04-03
> >
> > The authors have addressed my main concerns well, and most of the issues I raised have been largely resolved. The rebuttal is clear and helpful.

---

### Decision · Program_Chairs · 2026-04-30

**Decision:**

Accept (regular)

**Comment:**

This paper addresses humanoid motion tracking and provides a compelling investigation into the role of data quality versus quantity. The authors propose a systematic pipeline for filtering motions with low quality, low diversity, and limited dynamics. The empirical results are particularly strong: models trained on only 3%–10% of the full dataset are shown to outperform those trained on the complete dataset.

The reviewers were unanimous in recommending acceptance. They found the paper technically solid, practically significant, and well-validated experimentally. Overall, the contribution is well recognized and positively received by all reviewers.